# Experiences from six years of quality assured Model of End Stage Liver Disease (MELD) diagnostics

Pascal Hunold[1], Thomas Berg[2], Daniel Seehofer[3], Robert Sucher[3], Adam Herber[2], Berend Isermann[1], Joachim Thiery[1], Thorsten Kaiser[1]*

1 Institute of Laboratory Medicine, Clinical Chemistry and Molecular Diagnostics, University Hospital Leipzig, Leipzig, Germany, 2 Clinic and Polyclinic for Gastroenterology, Hepatology, Infectious Diseases and Pneumology, Division of Hepatology, University Hospital Leipzig, Leipzig, Germany, 3 Department of Visceral, Transplant, Thoracic and Vascular Surgery, University Hospital Leipzig, Leipzig, Germany

* thorsten.kaiser@medizin.uni-leipzig.de

## Abstract

### Background

The model of end-stage liver disease (MELD) score was established for the allocation of liver transplants. The score is based on the medical laboratory parameters: bilirubin, creatinine and the international normalized ratio (INR). A verification algorithm for the laboratory MELD diagnostic was established, and the results from the first six years were analyzed.

### Methods

We systematically investigated the validity of 7,270 MELD scores during a six-year period. The MELD score was electronically requested by the clinical physician using the laboratory system and calculated and specifically validated by the laboratory physician in the context of previous and additional diagnostics.

### Results

In 2.7% (193 of 7,270) of the cases, MELD diagnostics did not fulfill the specified quality criteria. After consultation with the sender, 2.0% (145) of the MELD scores remained invalid for different reasons and could not be reported to the transplant organization. No cases of deliberate misreporting were identified. In 34 cases the dialysis status had to be corrected and there were 24 cases of oral anticoagulation with impact on MELD diagnostics.

### Conclusion

Our verification algorithm for MELD diagnostics effectively prevented invalid MELD results and could be adopted by transplant centers to prevent diagnostic errors with possible adverse effects on organ allocation.

**Data Availability Statement:** Due to legal restrictions, (EU General Data Protection Regulation, Federal Data Protection Act (Germany), sharing of de-identified data sets is restricted. Our

data sets include personal patient's information such as their individual patient-ID, day of birth, case-ID and administrational- ID. Even though no clear names are included, an explicit identification may be possible. However, we invite other researchers to receive our data for reevaluation or further investigation of subsets of our data or the whole data after signing a contract according to the General Data Protection Regulation (GDPR) (https://gdpr-info.eu/art-28-gdpr/). Please contact the corresponding author in these cases. Additionally, the Institute of Laboratory Medicine, Clinical Chemistry and Molecular Diagnostics of the University Hospital Leipzig (E-Mail: mb-sekilm@medizin.uni-leipzig.de) or Prof. Dr.med. Berend Isermann (E-Mail: Berend.Isermann@medizin.uni-leipzig.de) can be contacted in these cases.

**Funding:** We acknowledge Leipzig University for the Open Access Publishing fund, and for providing partial support for the publication costs for this paper. The authors received no specific funding for conducting this work.

**Competing interests:** The authors have declared that no competing interests exist.

## Introduction

End stage chronic liver and metabolic diseases can only be effectively treated with orthotopic liver transplantation [1]. The allocation of liver grafts in many European and other countries is performed according to the laboratory model for end-stage liver disease (Lab-MELD) score. The Lab-MELD score is calculated based on three blood biomarkers: bilirubin, creatinine and the international normalized ratio (INR) [2]. The MELD score predicts the three-month mortality rate for patients with end-stage liver disease.

Due to the importance of the MELD score for the further therapy options of the patient, a high validity of the diagnosis is essential. There are known limitations of the MELD score, which are further engraved by preanalytic limitations [3–5].

Dialysis influences the serum creatine concentration. Therefore, for calculation of the MELD score the creatine is set as 353.6 μmol/l (4.0 mg/dl), if at least two dialyses were performed within the prior week or if a patient received 24 hours of continuous veno-venous hemodialysis (CVVHD) within the prior week [6].

The lab-MELD score replaced the previously applied, more clinically-based organ allocation system in Germany in 2006 to reduce the mortality among patients on the waiting list [2, 7–11]. Countries in the Eurotransplant region (such as Austria, Belgium, Croatia, Germany, Hungary, the Netherlands, Luxembourg and Slovenia) use the MELD score for the allocation of organs for elective patients. The United Network for Organ Sharing, responsible for organ allocation in the United States of America, uses the MELD Na score for patients with an initial MELD score of at least 11 for allocation [12]. In addition to the three MELD parameters, since 2016 the sodium concentration in serum is included in the MELD Na score [13].

For a minority of the patients on the waiting list—for example, patients with hepatocellular carcinoma or primary sclerosing cholangitis and special complications—the transplantation guidelines define specific exceptions. In these cases the lab MELD score does not reflect the patient's urgency for liver transplant sufficiently. If the patient is eligible for standard exception, an individual match-MELD score under consideration of the waiting time is calculated and replaces the lab-MELD score for the allocation process. The clinical physician has to evaluate the criteria for standard exception according to disease- and country specific regulations [6, 7].

In 2012, irregularities in MELD diagnostics were reported in different transplant centers in Germany. All German liver transplant centers were reviewed by an external group of specialists from the German Medical Association. Four of the 46 centers failed with serious policy violations; for example, dialysis was stated but not performed, or blood samples were manipulated. These violations had the consequence that for some patients on the waiting list, priority for liver transplantation was falsely increased [14]. These violations contributed to a decrease in organ donation in Germany. As a potential result, the shortage of donor organs was further aggravated [15, 16].

To improve the validity of the MELD diagnostics and to prevent irregularities, we implemented a specific quality-ensured lab-MELD diagnostic in 2012 (Fig 1). This study aimed to prove the effectiveness of our quality assurance procedure to avoid misclassifications of patients with indication for a liver transplant.

Conspicuous or implausible constellations may include: deviation in comparison to previous analysis; improbable changes in creatine concentration, bilirubin concentration, or INR values; and preanalytical errors such as hemolysis, or extended time spans between different components of the same order or between test ordering and arrival at the laboratory. Implausible constellations are checked by personnel specifically trained in laboratory medicine in the context of liver synthesis parameters such as fibrinogen, albumin or the choline esterase rate.

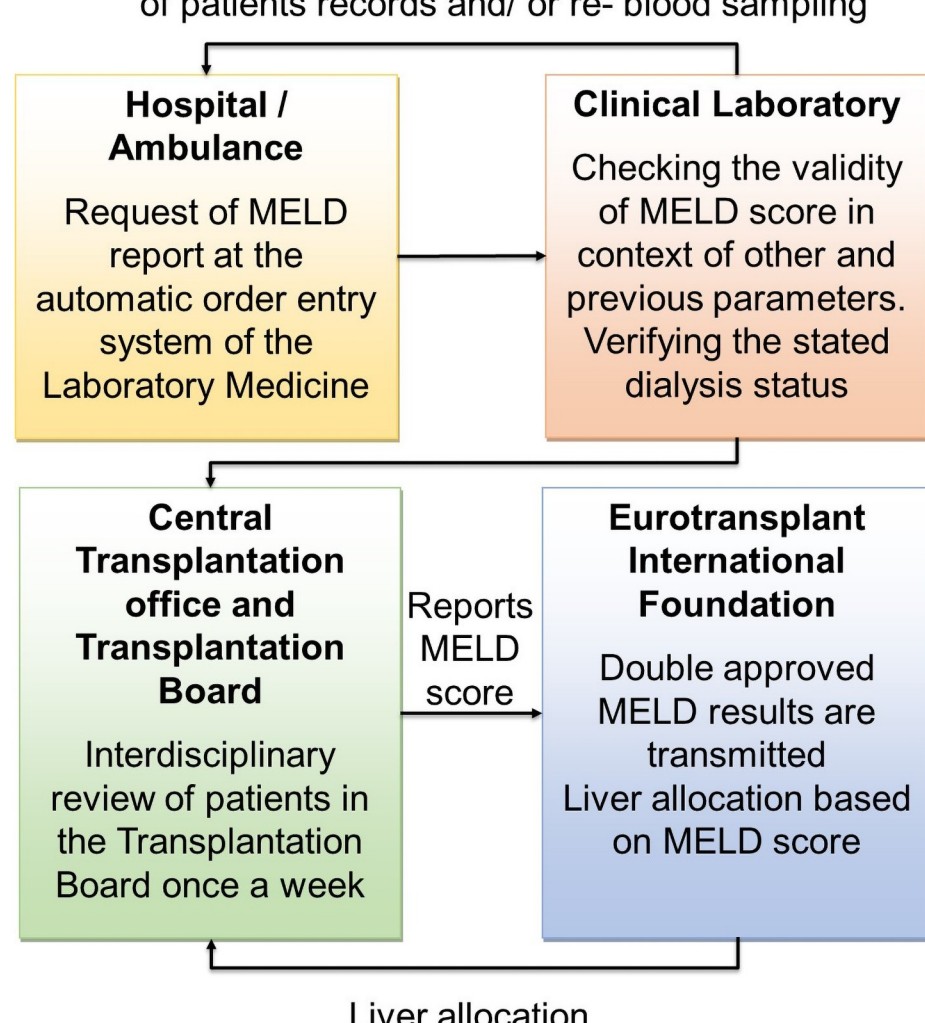

**Fig 1. Flow sheet of quality approved lab-MELD diagnostic at the University Hospital Leipzig.**

## Methods

The lab-MELD validation and reporting system (Fig 1) was implemented as a verification algorithm in the laboratory information system in November 2012 by the Institute of Laboratory Medicine. Before the implementation of the specific quality assurance of MELD diagnostics, there was an interdisciplinary discussion at the University Hospital Leipzig. The implementation was unanimously decided by the Institute of Laboratory Medicine; the Department of Visceral, Transplantation-, Vascular and Thoracic Surgery; the Section of Hepatology; the Department of Gastroenterology and Rheumatology; and the Hospital Management Board.

The Institute for laboratory medicine, as an independent institution, supervised the preanalytical and clinical conditions and the plausibility of the results for every patient (Fig 1).

The MELD diagnostic was requested electronically by the clinicians using the laboratory information system. This request also included the specification of hemodialysis and anticoagulation status.

The MELD parameters had to be ordered within the same laboratory order and needed to be send in at the same time. During the process of medical validation, the MELD score and its components were specifically screened for preanalytical errors, such as the plausibility of the dialysis status and the anticoagulation status, misidentification of patients, prolonged preanalytical phase and other typical preanalytical errors.

After the process of validation, the MELD parameters, as well as the MELD score and the dialysis status were reported in a specific document for further transfer of the data to Eurotransplant. Eurotransplant is responsible for the allocation of donor organs.

Lab-MELD diagnostic was performed when indicated for clinical diagnosis of patients in evaluation for orthotopic liver transplantation. A notification during the order creation ensured the use for this indication. There was no need for additional blood draws for this study beyond those needed for MELD score determination. If necessary, additional clinical information had been collected from the electronic patient records. Bilirubin was measured photometrically using the test kit "Bilirubin Total Gen.3" on Cobas8000 c701. Creatine was measured photometrically, enzymatically using the test kit "Creatine plus ver.2" on Cobas 8000 c701. INR is calculated parameter based on the prothrombin time measured on ACL as coagulometry using the testkit "HaemosIL Thrombin Time". All methods were carried out in accordance with applicable guidelines and regulations. The institute of Laboratory Medicine is accredited according to DIN EN ISO 15 189:2014. For measurements the regulations of the Guideline of the German Medical Association on Quality Assurance in Medical Laboratory Examinations (Rili-BAEK) were assured and certified within external quality assessment (EQA) [17].

The study approach was approved by the Faculty of Medicine's ethics committee of the Leipzig University.

The diagnostic workup of the MELD diagnostic required quality assurance of the clinical diagnostic process and ethics committee agreed to the usage of these results also for scientific investigation without the specific informed consent of the patients.

All MELD-requests from November 2012 to October 2018 were included in the analysis. Only for analysis necessary patient information (day of birth, Patient-ID, Case-ID, Administrational ID) and information of the laboratory order (order-ID, all requested parameters) were accessed.

## Statistics

The data for all three analytes—serum bilirubin, serum creatinine and the INR—and the resulting MELD score were analyzed by mean values, standard deviations and the Chi-squared test using IBM SPSS Statistics for Windows (Version 24.0. Armonk, NY) and Microsoft Excel 2010 (Microsoft Corporation, Redmond, USA).

## Results

From November 2012 to October 2018, 7,270 lab-MELD (3.3 per day or 23.4 per week) diagnostics were performed. Table 1 summarizes the characteristics of cases of 1,494 different patients that required MELD diagnostics.

### Specific medical laboratory validation of MELD diagnostics

During the first six years, 7,270 MELD diagnostics were performed. In 97.3% of the cases (7,077), a valid MELD score could be reported. In total 524 patients received anticoagulation treatment. Of those, 2 patients were treated with Fondaparinux, 488 with heparin, 33 with a Vitamin K antagonist and 1 patient with a suggested intake of Apixaban, a direct oral FXa

**Table 1.  Summary of the characteristics of cases that required MELD diagnostic from November 2012 until October 2018 within the university hospital Leipzig.**

| | Women | Men | Total | p–Value |
|---|---|---|---|---|
| **MELD- diagnostics (number of different patients)** | 2633 (553) | 4637 (941) | 7270 (1494) | |
| **Age (Range) [years]** | 55.0 (5–81) | 55,7 (1–84) | 55.5(1–84) | |
| **Bilirubin (Range) [µmol/L]** | 34.8 (2.2–1096,8) | 31.6 (2.6–785.4) | 31.7 (2.2–1096.8) | 0.006 |
| | n = 2625 | n = 4627 | n = 7252 | |
| **Creatinin (Range) [µmol/L]** | 81 (20–610) | 89 (21–786) | 86 (20–786) | < 0.001 |
| | n = 2628 | n = 4628 | n = 7256 | |
| **INR (Range)** | 1.4 (0.8–9) | 1.3 (0.8–12.9) | 1.3 (0.8–12.9) | < 0.001 |
| | n = 2589 | n = 4583 | n = 7172 | |
| **Dialysis** | 253 (9.7%) | 305 (6.6%) | 558 (7.7%) | < 0.001 |

Only valid results that could be reported to the transplant organization are displayed.

inhibitor). These MELD scores were separately validated for plausibility by comparing the current results to previous results without anticoagulation in the clinical context and INR was unchanged.

In the remaining 193 cases the initially requested MELD score could neither be validated nor reported to the transplant organization for different reasons. These cases are illustrated in detail in Fig 2.

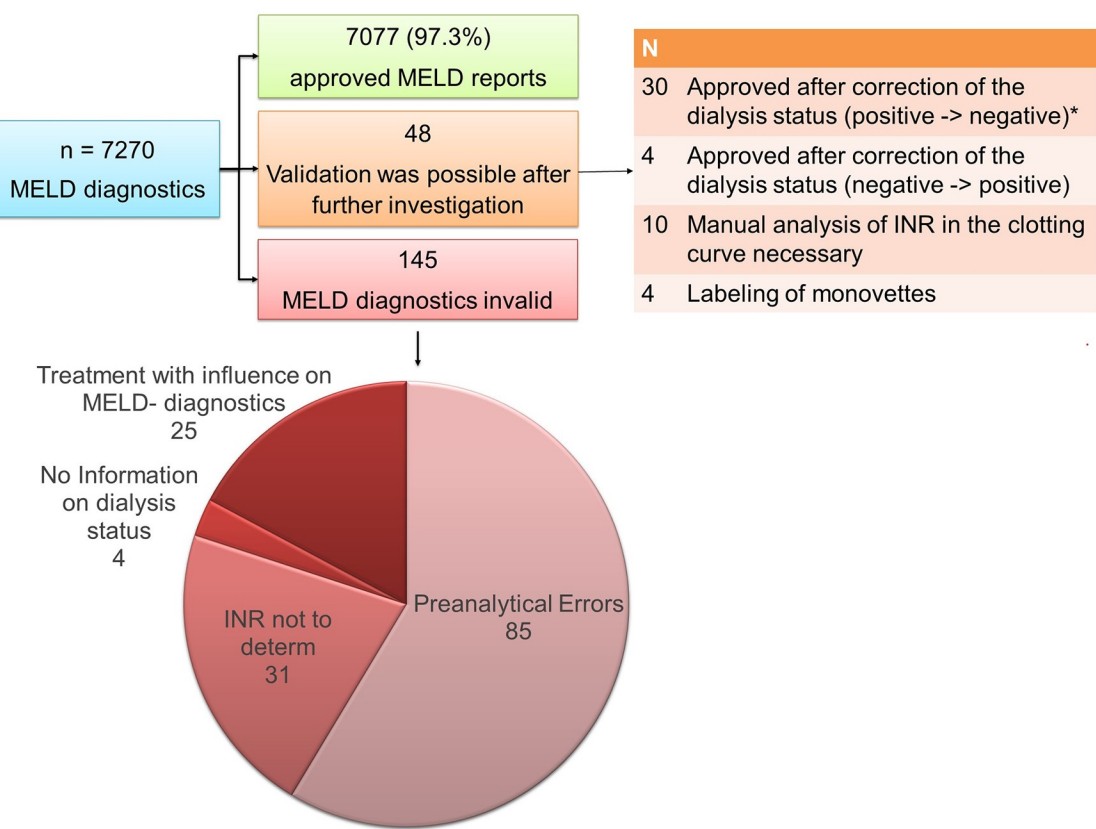

**Fig 2. Summary of the requested MELD scores and the results of the laboratory medical validation.** Anticoagulation might be indicated within patients with liver diseases. However, INR must be unchanged for valid MELD- diagnostics. In 25 cases, treatment with anticoagulation had influence on the INR and consequently on the MELD score. Therefore, a validation was not possible. * Dialysis status or interval did not fulfill the official criteria. However, no cases of deliberate misreporting were identified.

After individual investigation, 48 valid MELD scores could be provided. In 30 of these cases, the indicated positive dialysis status could not be confirmed by checking the patient's record and consulting the sender of the samples. In 8 cases dialysis was performed; however, but not as defined in the dialysis criteria within the week; CCVHD or CVHD were not performed for the demanded time [6]. For 22 cases the lab data were not congruous with performance of dialysis. After consultation with the clinical physicians a mistake during order creation was evaluated. One patient received Albumin dialysis. This method does not equal a kidney replacement method. Dialysis was changed in these 30 cases to "no dialysis" resulting in lower MELD scores (median 33.0 [before] vs. 22.5 [after]). After this correction, the results were approved and reported to the transplant organization.

In four cases, the dialysis status was incorrectly indicated as negative when the diagnostic was requested. After consulting the clinics and the dialysis unit, the status was corrected to "yes".

In the remaining 145 cases, the MELD score could not be reported at all to the transplant organization due to different reasons (Fig 2).

In 85 cases, the MELD score could not be determined or reported to the transplant office because of major preanalytical errors with a probable impact on the resulting MELD score (Table 2).

In these cases, a new blood sample had to be taken to guarantee a valid MELD score. In four cases, the sender did not state the status of dialysis and the status could not be ascertained. In seven specimens, the citrate tubes were significantly underfilled with a probable impact on INR and were not usable for a valid MELD diagnostic, as underfilling can lead to false high INR values. In seven cases, there was a confusion of patients' identity during blood withdrawal, which was identified during medical validation. In five cases, a MELD score was requested for children aged 1–6 years but was not reported as the MELD score is only validated for adolescents and adults aged 12 years and above. In these cases, the clinical physician was informed and the pediatric end-stage liver disease (PELD) score was used [6]. Since PELD diagnostic is only rarely requested at our laboratory and has specific requirements and exceptions, it was not implemented in our verification algorithm [6].

In 24 cases with anticoagulation, there was a significant influence on INR and consequently the MELD score. In these cases, the INR (in median 2.65) and MELD scores (in median 21.0) were significantly higher compared to the rest of the patients (p < 0.001 and p = 0.014, respectively). Seventeen of these cases were due to the intake of Vitamin K antagonists and were already stated in the MELD diagnostic request. In the process of validation by the laboratory, one case of the intake of Vitamin K antagonists was unraveled, which was not indicated before. Another four cases of Rivaroxaban intake, one of Apixaban intake (both direct oral factor Xa-inhibitors), one other not specified oral antagonist and one case of a massive overdose of

**Table 2. Table with preanalytical errors leading to not reporting MELD score.**

| Preanalytical errors leading to invalid MELD diagnostic | |
|---|---|
| Material error (missing material (citrate blood tube or serum tube missing), material-ID did not match tube, lithium heparin/ascites instead of blood serum) | 56 |
| Underfilling with inadequate mixing ratio citrate vs. blood | 7 |
| Misidentification of patients in the process of blood withdrawal (wrong ID on the monovette) | 7 |
| Prolonged preanalytical period (>4h) | 5 |
| MELD requested for children under the age of 12 years (PELD necessary) | 5 |
| Citrate blood clotted | 3 |
| Late laboratory entrance | 2 |

heparin, which relevantly inhibited the prothrombin time test, was identified by the laboratory and could therefore not be reported.

## Invalid clotting tests in patients who required MELD diagnostics

In 42 of the 7,270 (0.57%) cases, the automatized clotting tests did not display results because of the massive lack of coagulation factors in advanced liver failure. In 10 cases, a result could be achieved through manual interpretation of the clotting curve by the laboratory physician. The INR in these patients was between 3.3 and 9 with a resulting MELD score from 27 to 40. In one of these cases, the INR result was not plausible in the context of the pre-results for unknown reasons. The follow-up measurements of the patient resulted in plausible results.

In 31 specimens of 20 patients, a valid determination of the prothrombin time/INR was not possible. In these cases, an additional sample was requested, which was sent in in 24 cases. In 16 of these cases, a prothrombin time/INR diagnostic was possible in the median after 23.9 h (4.4–48.2 h). Ten of these cases had received a prior substitution of coagulation factors or fresh frozen plasma.

Importantly, 13 of these 21 patients (62%) died within 2 days (0–88 days) after the first undeterminable prothrombin time/INR and 4 patients received liver transplantation within 6.5 days (0–16 days).

## Discussion

We report for the first time experiences with a specific laboratory quality assurance system to improve the validity of MELD diagnostic. To the best of our knowledge, our transplant center was the first to implement such a specific verification algorithm for MELD diagnostics in the clinical routine.

The majority of the MELD diagnostics (97.3%) passed the process of analysis and validation without any doubt of validity. However, the MELD diagnostic can be complex as many regulatory requirements exist. In this study, we identified different deviations from valid MELD diagnostics.

Our verification algorithm does not absolve clinicians from meaningful indication, outcome assessment, and knowledge of clinical exceptions. The clinicians also have to be aware of the follow-up examinations depending on the MELD score and in compliance with the Eurotransplant regulations [6]. For various diseases the lab MELD score does not reflect the patient's urgency for liver transplant sufficiently. In these cases, the patient is eligible for standard exception (SE) and the match MELD score will be used. The criteria for SE are disease and country specific. The clinical physicians have to be aware of all the applicable regulations [6]. The basis of the match MELD score is the clinical evaluation and diagnostics of the clinical physicians. The match MELD score is not incorporated in our system and is subject to the patient´s specific discussion in the interdisciplinary liver transplant conference.

The ordering physicians stated the dialysis status during the laboratory order creation. The status was checked for plausibility in the context of the other laboratory results during medical validation. However, in some cases, the stated dialysis status did not fulfill the official criteria. Importantly, not knowing the relevant patient's characteristics, the unawareness of the regulations as well as inadvertent mistakes during order creation could lead to an incorrect stated status. After medical validation by the laboratory physician, no cases of deliberate misreporting were identified. The transplantation office reconfirmed each of the cases if the dialysis status met the official dialysis criteria by checking the dialysis protocols before the admittance to the transplant organization.

The intake of oral anticoagulation also influences the MELD score by raising the INR [18–20]. The deliberate or unintentional concealment of the intake of these drugs could lead to elevated MELD scores and the prioritization of those patients in the process of allocation. During the medical validation, the stated anticoagulation and the INR were validated in the context of other biomarkers for liver function, such as albumin, cholinesterase and bilirubin. Furthermore, the INR was compared to previous results and analyzed for implausible constellations. INR results indicating the use of anticoagulation were not reported. However, from the laboratory parameters alone, it is not possible to exclude the use of anticoagulation in patients with minor changes in INR.

A very relevant and not conclusively solved problem is the handling of undeterminable INR in the majority of cases due to the lack of fibrinogen. In the case of an extreme lack of coagulation factors, the prothrombin time and the calculated INR are dependent on the time used to determine the clotting reaction. There were 21 patients with no clotting within 315 sec. This maximum waiting time for the coagulation reaction to occur is individually defined by the medical laboratories. In the case of a tremendous lack of fibrinogen, independently from other clotting factors concentrations, no reaction will be detectable at all and the laboratory calculates an INR based on the calibration curve at the maximum of the waiting time. The calculation of the INR and the submission to the transplant organization with a longer waiting time results in extremely high MELD scores and increased chances to receive a transplant. The objectives of Eurotransplant state that in these cases, an Eurotransplant liver intestine advisory committee (ELIAC) auditor could determine an INR equivalent using the highest prothrombin time from the conversion table provided by the laboratory since the tables differ among the laboratories and manufacturers [6]. In our experience, the procedure is inconsistent and needs further clarification of the regulations for Eurotransplant as well as United Network for Organ Sharing (UNOS). For unknown reasons, the advisory committee decided only to accept the last determinable INR that resulted in a lower MELD score compared to the results obtained from the maximum of the INR-conversion table. This is highly relevant, as many patients with an undeterminable INR had to be supplemented with clotting factors and died within a short time (13 of 21 patients [62%] in our study).

Most problems leading to an invalid result in laboratory measurements occurred in the context of the preanalytical phase [21]. The medical validation of the MELD score could help to reveal and reduce the problems in the preanalytic phase of the diagnostic process. The confusion of patients' identity is a major problem leading to inaccurate MELD scores. We identified seven cases.

Another relevant preanalytical problem could be time-dependent changes in the specimen. The delayed laboratory analysis of citrate blood could influence the INR leading to a falsely elevated MELD score [22]. In our study, no hints of outdated samples were detected.

Another limitation of MELD Diagnostics cannot be solved by our system. There is already evidence that the measurement methods have an impact on the concentration of the analytes and the MELD score [23–27]. Therefore, methods used for the determination of the parameters needed to calculate the MELD score should be specified to guarantee an equitable allocation.

As MELD diagnostics are of life-determining consequences, a valid determination needs to be assured. We were able to show the positive effects of specific quality assurance of MELD diagnostics. Advantages of our approach are an identification and prevention of invalid MELD diagnostics in a relevant number of cases with invalid dialysis status, impact of anticoagulation therapy and further preanalytical errors. Policy violations as occurred in German liver transplant centers in 2012 could have been revealed with very high probability using our specific verification algorithm for MELD diagnostics.

Furthermore, our system allows to preserve blood samples for quality assurance and remeasurements if indicated. A broad implementation of a laboratory quality assurance system in transplant centers could help to guarantee that organ allocation is directly related to the severity of the liver disease.

## Author Contributions

**Conceptualization:** Pascal Hunold, Thomas Berg, Daniel Seehofer, Robert Sucher, Adam Herber, Berend Isermann, Joachim Thiery, Thorsten Kaiser.

**Formal analysis:** Pascal Hunold, Thomas Berg, Daniel Seehofer, Berend Isermann, Joachim Thiery, Thorsten Kaiser.

**Investigation:** Pascal Hunold.

**Methodology:** Pascal Hunold, Thorsten Kaiser.

**Project administration:** Thorsten Kaiser.

**Validation:** Thorsten Kaiser.

**Visualization:** Pascal Hunold, Thorsten Kaiser.

**Writing – original draft:** Pascal Hunold, Thorsten Kaiser.

**Writing – review & editing:** Pascal Hunold, Thomas Berg, Daniel Seehofer, Berend Isermann, Joachim Thiery, Thorsten Kaiser.

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
