## [Decision Letter · Decision Letter 0]

19 Jan 2021

PONE-D-20-34374

Experiences from Six Years of Quality Assured Model of End Stage Liver Disease (MELD) Diagnostics

PLOS ONE

Dear Dr. Kaiser,

Thank you for submitting your manuscript to PLOS ONE. After careful consideration, we feel that it has merit but does not fully meet PLOS ONE’s publication criteria as it currently stands. Therefore, we invite you to submit a revised version of the manuscript that addresses the points raised during the review process.

We look forward to receiving your revised manuscript.

Kind regards,

Pal Bela Szecsi, M.D. D.M.Sci.

Academic Editor

PLOS ONE

Journal Requirements:

2.We note that you have indicated that data from this study are available upon request. PLOS only allows data to be available upon request if there are legal or ethical restrictions on sharing data publicly. For information on unacceptable data access restrictions, please see http://journals.plos.org/plosone/s/data-availability#loc-unacceptable-data-access-restrictions.

3.Thank you for stating the following financial disclosure:

4. Please ensure you have discussed any potential limitations of your study in the Discussion.

5. In your Methods section, please provide additional information about the medical data/samples collected and the demographic details of the human subjects. Please ensure you have provided sufficient details to replicated the analyses such as: a) a description of any inclusion/exclusion criteria that were applied to the medical records/samples, b) a table of relevant demographic details c) a description of how the medical records/samples were collected.

6. In the ethics statement in the manuscript and in the online submission form, please provide additional information about the patient records/samples used in your retrospective study, including: a) whether all data were fully anonymized before you accessed them; b) the date range (month and year) during which patients' medical records/samples were accessed.

Additional Editor Comments:

I find the manuscript of some value, even though one reviewer recommend rejection.

Anyhow, I’ll allow a major revision.

You should emphasize more on the laboratory errors around reporting the MELD score.

Being in a laboratory myself, I wonder why you system does not automatically request all three parameters when MELD is ordered and why it not automatically substitute MELD with PELD when the age is <12yrs?

In table 2, several of the errors seems belonging to the same group, missing material, wrong material, lithium instead of serum. “Confusion of patients in the process of blood withdrawal” is that the code for incorrect ID? “Incorrect labeling of monovettes (wrong material)” is that patient label og a producer error?

Some details concerning the analytic methods would be nice, calibration CV, QC.

SPSS is not originating in Ehningen, the correct citation should be “IBM Corp. Released 2016. IBM SPSS Statistics for Windows, Version 24.0. Armonk, NY: IBM Corp”, or some abbreviation hereof.

Reviewers' comments:

Reviewer's Responses to Questions

**Comments to the Author**

1. Is the manuscript technically sound, and do the data support the conclusions?

Reviewer #1: Partly

Reviewer #2: Yes

2. Has the statistical analysis been performed appropriately and rigorously? 

Reviewer #1: Yes

Reviewer #2: Yes

3. Have the authors made all data underlying the findings in their manuscript fully available?

Reviewer #1: Yes

Reviewer #2: No

4. Is the manuscript presented in an intelligible fashion and written in standard English?

Reviewer #1: Yes

Reviewer #2: Yes

5. Review Comments to the Author

Reviewer #1: The paper is reviewing the experience of a quality assurance system of validation and report of MELD score. All the results shows the good performance of this system to detect invalid MELD scores due to different reasons .

I think these results should be submitted to a different type of journal more focused in Clinical Chemistry and/or Laboratory

In the discussion , it should be emphasized to interpret this MELD score in the clinical context of patients . The clinician must understand , before sending the data , the intrinsic limitations of MELD . This score is NOT design for patient under dialysis or under anticoagulation therapy .

All this scores must be validated before and/or after the results of the laboratory to the final use of allocation of patients in a waiting list . There is a some exceptions for MELD and also some limitations that make essential the evaluation of the physician. All this aspects must be discussed in the paper

Reviewer #2: The significance of this study, and what exactly was accomplished, needs to be more strongly justified. The introduction should provide more clarification as to what kinds of laboratory errors this diagnostic review algorithm is supposed to address. Additionally, while a recent laboratory validation controversy in Germany is mentioned, further support for the need for a validation algorithm would strengthen the study. The manuscript as a whole also would benefit to referring to the "laboratory diagnostic" as a verification algorithm, or some similar terminology, because as it is written, it initially appeared that the study was on a new form of laboratory test that somehow calculates MELD directly from a sample.

A more granular and clearly labeled diagram of the verification algorithm should be included. It is hard to tell what types of findings would trigger review by the laboratory medicine group in this "diagnostic", and therefore these findings are, as yet, not applicable to other centers. As this is an implementation/quality improvement type project, a clear algorithm would be helpful, or at least examples: for example, where you state that "conspicuous constellations" in Figure 1 trigger a review, suggest some of these conspicuous constellations.

Page 11, line 105: "blood takings" should be "blood draws" or "blood samples needed to be procured". Line 111 on the same page should have "the" before "ethics committee". Note that in line 163, you state there were cases in which further blood draws were performed if the original sample was inadequate; please change line 105's statement to reflect that this study did not require further blood draws beyond those needed for adequate MELD determination.

6. PLOS authors have the option to publish the peer review history of their article (what does this mean?). If published, this will include your full peer review and any attached files.

Reviewer #1: No

Reviewer #2: No

---

## [Author Response · Author response to Decision Letter 0]

23 Feb 2021

We found the reviewer’s comments and suggestions helpful and revised our manuscript accordingly. 

Please find our responds embedded within the “Response to reviewers”. 

Our responses to each comment is in bold print. 

The corresponding changes are highlighted in the manuscript.

23.02.2021: We added additional contact information for data access requests in the revised cover letter.

---

## [Decision Letter · Decision Letter 1]

21 May 2021

PONE-D-20-34374R1

Experiences from Six Years of Quality Assured Model of End Stage Liver Disease (MELD) Diagnostics

PLOS ONE

Dear Dr. Kaiser,

Thank you for submitting your manuscript to PLOS ONE. After careful consideration, we feel that it has merit but does not fully meet PLOS ONE’s publication criteria as it currently stands. Therefore, we invite you to submit a revised version of the manuscript that addresses the points raised during the review process.

A few issues still need to be addressed. Please correct and the MS will shortly be accepted.

We look forward to receiving your revised manuscript.

Kind regards,

Pal Bela Szecsi, M.D. D.M.Sci.

Academic Editor

PLOS ONE

Journal Requirements:

Reviewers' comments:

Reviewer's Responses to Questions

**Comments to the Author**

1. If the authors have adequately addressed your comments raised in a previous round of review and you feel that this manuscript is now acceptable for publication, you may indicate that here to bypass the “Comments to the Author” section, enter your conflict of interest statement in the “Confidential to Editor” section, and submit your "Accept" recommendation.

Reviewer #2: (No Response)

Reviewer #3: All comments have been addressed

2. Is the manuscript technically sound, and do the data support the conclusions?

Reviewer #2: Yes

Reviewer #3: Yes

3. Has the statistical analysis been performed appropriately and rigorously? 

Reviewer #2: Yes

Reviewer #3: Yes

4. Have the authors made all data underlying the findings in their manuscript fully available?

Reviewer #2: Yes

Reviewer #3: Yes

5. Is the manuscript presented in an intelligible fashion and written in standard English?

Reviewer #2: Yes

Reviewer #3: Yes

6. Review Comments to the Author

Reviewer #2: Line 22: add “the” before “laboratory MELD diagnostic” or make diagnostics plural, for appropriate grammar

Lines 78-79: correct conventions such as follows: “policy violations; for example, dialysis was stated but not performed, or blood samples were manipulated.”

Lines 79-81: correct conventions such as follows: “These violations had the consequence that for some patients on the waiting list, priority for liver transplantation was falsely increased.”

Lines 90 to 96: Change “constellations may be” to “may include”; change “remarkably” to a more concrete term such as “improbable” or “extreme.” Also, what does “materials of the same order” mean? Do you mean “components”? Change “specifical” to “specifically”. Also, use semi-colons instead of commas to separate list items when list items contain a list – i.e., “deviation in comparison to previous analysis; improbable changes in creatine, bilirubin concentration, or INR values; and preanalytical errors such as hemolysis, or extended time spans between different materials of the same order or between test ordering and arrival at the laboratory.”

Lines 116-117: “confusion of patients” should be restated. I believe you are referring to misidentification of patients – rather than confusion as a mental status.

Lines 123-124: Restate “A notification during the order creation informed ensured the use for this indication.” It appears informed is an extra word here.

Table 2: Does “missing material” refer to a missing blood draw, an incorrect tube used, or something else? Again, as above, use “misidentification” rather than “confusion” to refer to incorrect patient ID

Lines 258-259: Add “the” before “basis”

Line 260: replace the word “embraced” with something more clinical such as “incorporated”; add “a” or “the” before “patient’s specific”

Line 265: add “the” before “relevant patient’s”

Reviewer #3: This is a revised version of a paper that I did not originally review. The information provided is of interest. The authors have addressed all previous comments in a satisfactory way.

7. PLOS authors have the option to publish the peer review history of their article (what does this mean?). If published, this will include your full peer review and any attached files.

Reviewer #2: No

Reviewer #3: No

---

## [Author Response · Author response to Decision Letter 1]

1 Jun 2021

Thank you for the review. Please find our responses to your comments and review within the rebuttal letter. Thank you!

---

## [Decision Letter · Decision Letter 2]

23 Jun 2021

Experiences from Six Years of Quality Assured Model of End Stage Liver Disease (MELD) Diagnostics

PONE-D-20-34374R2

Dear Dr. Kaiser,

We’re pleased to inform you that your manuscript has been judged scientifically suitable for publication and will be formally accepted for publication once it meets all outstanding technical requirements.

Kind regards,

Pal Bela Szecsi, M.D. D.M.Sci.

Academic Editor

PLOS ONE

Additional Editor Comments (optional):

Reviewers' comments:

Reviewer's Responses to Questions

**Comments to the Author**

1. If the authors have adequately addressed your comments raised in a previous round of review and you feel that this manuscript is now acceptable for publication, you may indicate that here to bypass the “Comments to the Author” section, enter your conflict of interest statement in the “Confidential to Editor” section, and submit your "Accept" recommendation.

Reviewer #2: All comments have been addressed

2. Is the manuscript technically sound, and do the data support the conclusions?

Reviewer #2: Yes

3. Has the statistical analysis been performed appropriately and rigorously? 

Reviewer #2: Yes

4. Have the authors made all data underlying the findings in their manuscript fully available?

Reviewer #2: No

5. Is the manuscript presented in an intelligible fashion and written in standard English?

Reviewer #2: Yes

6. Review Comments to the Author

Reviewer #2: All comments from this revision pertained to grammar and conventions, which have all been satisfactorily addressed.

7. PLOS authors have the option to publish the peer review history of their article (what does this mean?). If published, this will include your full peer review and any attached files.

Reviewer #2: No

---

## [Editor Report · Acceptance letter]

18 Aug 2021

PONE-D-20-34374R2 

Experiences from six years of Quality Assured Model of End Stage Liver Disease (MELD) Diagnostics 

Dear Dr. Kaiser:

I'm pleased to inform you that your manuscript has been deemed suitable for publication in PLOS ONE. Congratulations! Your manuscript is now with our production department. 

Kind regards, 

on behalf of

Dr. Pal Bela Szecsi 

Academic Editor

PLOS ONE